# YOLOv6: A Single-Stage Object Detection Framework for Industrial Applications

## Abstract

We inaugurate a new state-of-the-art real-time object detector YOLOv6, which comprises a series of novel hardware-aware architectures accompanied by a set of unique training schemes tailored for industrial scenarios. For a glimpse of performance, our YOLOv6-N hits 37.5% AP on the COCO dataset at a throughput of 1187 FPS tested with an NVIDIA Tesla T4 GPU. YOLOv6-S strikes 45.0% AP at 484 FPS, outperforming other mainstream detectors at the same scale (YOLOv5-S, YOLOv8-S, YOLOX-S and PPYOLOE-S). Meantime, YOLOv6-M and L achieve better accuracy performance (50.0%/52.8% respectively) than other detectors at a similar inference speed. Additionally, with an extended backbone and neck design, our YOLOv6-L6 achieves the state-of-the-art accuracy in real-time object detection. We carefully conducted extensive experiments to validate the effectiveness of each proposed component.

## 1 Introduction

As a fundamental computer vision task, object detection has been studied in depth for decades. With the rise of deep learning, object detectors are designed to supply ideal accuracy with high inference speed, which forms a critical sub-task: *real-time object detection*. YOLO series (Redmon et al., 2016; Redmon & Farhadi, 2017; 2018; Bochkovskiy et al., 2020; Glenn, 2022; Wang et al., 2022; Glenn, 2023; Ge et al., 2021b; Xu et al., 2022) have been the most popular real-time detection frameworks in industrial applications. These detectors are carefully designed to have advanced network structures, training strategies, and loss functions in pursuit of an excellent balance between speed and accuracy. Recent years have witnessed the development of anchor-free detectors (Law & Deng, 2018; Tian et al., 2019; Zhou et al., 2019), which well-balances the accuracy and speed performance as well.

Nevertheless, on the one hand, rare attention has been paid to extreme throughputs at deployment, which causes a challenging bottleneck in handling ever-increasing data traffic. On the other hand, training techniques that improve detection performance without impacting inference speed have not been much investigated specifically for real-time object detection. In contrast, *self-distillation* (Zhang et al., 2019; Mobahi et al., 2020; Zhang & Sabuncu, 2020) has been studied for years to provide extra performance without introducing extra models and inference computation cost. Additionally, *auxiliary training* (Zhang et al., 2020a) equips the network with a set of auxiliary modules in training to integrate knowledge from various perspectives.

With the aforementioned observations in mind, we bring the birth of YOLOv6. We specifically design a series of powerful backbones that achieve excellent trade-offs in terms of throughput and performance with the help of reparameterization techniques (Ding et al., 2021). We also particularly devise an *adaptive self-distillation strategy* for the real-time object detector, in which the distillation process can be divided into different stages where the student dynamically adjusts the proportion of knowledge either from the hard labels or from the soft ones of the teacher. For large models, we leverage Distribution Focal Loss (DFL) (Li et al., 2020) as the regression loss for localization distillation. Whereas for small models, we invent *Decoupled Localization Distillation* (DLD), which adopts both a lightweight regression branch and another heavier regression branch for DFL in training. Only the lightweight one is retained after the training.

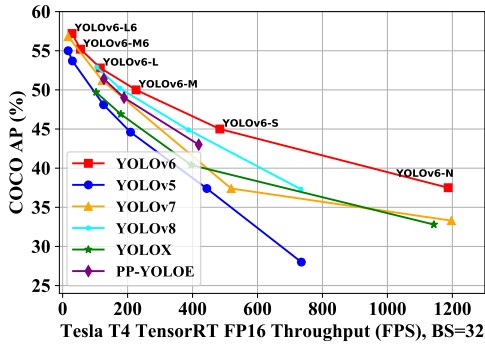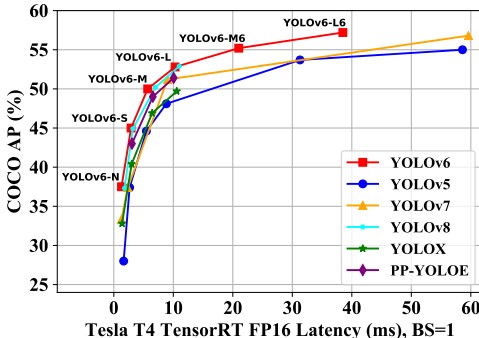

Figure 1: Comparison of state-of-the-art efficient object detectors. Both latency and throughput (at a batch size of 32) are given for a handy reference. All models are tested with TensorRT 7.

We analyzed the performance of detectors with either anchor-based or anchor-free paradigms and further proposed an *Anchor-aided Training* strategy to integrate the advantages of both two paradigms. Our contributions can be summarized as follows:

- We unveil a collection of architectures with different sizes tailored especially for industrial applications. The backbones are renovated with re-parameterization techniques. Branch choices (single/multiple) are made for models at different scales. We introduce a lightweight Bi-directional Concatenation (BiC) module and a SimCSPSPPF block to have a refurbished neck called RepBi-PAN, which brings significant performance gains.

- We propose an *anchor-aided training* (AAT) strategy to enjoy the advantages of both anchor-based and anchor-free paradigms without touching inference efficiency. A new adaptive self-distillation strategy is invented to further boost the performance. For smaller models, *Decoupled Localization Distillation* (DLD) is specifically schemed to avoid the notable speed decline.

- We achieve a new state of the art on the COCO dataset in terms of throughput and performance at all compared scales. The comparison is shown at a similar scale in Fig. 1, where YOLOv6 of all sizes outperform other competitors considering accuracy and speed.

## 2 RELATED WORK

**Real-time Object Detection** Object detectors can be categorized as two-stage or one-stage detectors. The representative methods of the former are R-CNN series (Girshick et al., 2014; Girshick, 2015; Ren et al., 2015). Real-time object detectors are usually one-stage methods providing faster inference speed. The most popular one-stage methods are the YOLO series, the pioneering works of which are YOLOv1-3 (Redmon et al., 2016; Redmon & Farhadi, 2017; 2018). They blaze a new trail of one-stage detectors along with the later substantial improvements. YOLOv4 (Bochkovskiy et al., 2020) reorganized the detection framework into several parts (backbone, neck, and head), and verified bag-of-freebies and bag-of-specials at the time to design a framework suitable for training on a single GPU. At present, YOLOv5 (Glenn, 2022), YOLOX (Ge et al., 2021b), PPYOLOE (Xu et al., 2022), YOLOv7 (Wang et al., 2022) and most recently YOLOv8 (Glenn, 2023) are all the competing candidates for efficient detectors to deploy.

Additionally, object detection methods are also classified into anchor-based or anchor-free detectors based on whether the pre-defined proposals are used. Pre-defined proposals were considered valid to improve the performance of detectors in early works of object detection (Ren et al., 2015; Redmon & Farhadi, 2017). However, recent works that abandon anchors (Law & Deng, 2018; Tian et al., 2019; Zhou et al., 2019) believe the hyperparameters brought by the anchors will impact the detection accuracy. And the well-designed anchor-free detectors bridge the performance gap between anchor-based detectors with even faster inference speed in the application.

**Auxiliary Training**  There are several prior works (Hao et al., 2020; Ma et al., 2021; RangiLyu, 2021; Wang et al., 2022) enhancing detection performance through auxiliary training without impacting inference speed. In (Hao et al., 2020), auxiliary supervision is introduced by mapping labels into latent embedding, which relies on an auxiliary module to encode ground-truth labels. IQDet (Ma et al., 2021) adds a Quality Distribution Encoder as an auxiliary subnet to sample high-quality training samples. The subnet is removed in the inference stage. NanoDet (Rangi-Lyu, 2021) uses the auxiliary branch to give a better label assignment strategy. The auxiliary head in YOLOv7 (Wang et al., 2022) is used to learn from the coarser labels providing additional supervision signals to enhance the learning capacity.

**Self-distillation**  Knowledge distillation is originally studied as a model compression technique (Hinton et al., 2015), which relies on an extra large teacher model to guide the learning of the small student model. With the development of knowledge distillation, relevant methods can be divided into three categories (Gou et al., 2021): (1) Response-based methods: the student model directly mimics the neural response of the last layer of the teacher model; (2) Feature-based methods: the outputs of teacher model's hidden layers are believed able to provide extra useful supervision signals; (3) Relation-based methods introduce relationship knowledge between different layers or data samples besides the responses and features. Most recently, knowledge distillation methods (Chen et al., 2017; Li et al., 2017; Wang et al., 2019; Dai et al., 2021; Guo et al., 2021; Zhixing et al., 2021; Zheng et al., 2022) are specifically designed for object detection.

Particularly, *self-distillation* stands out among all knowledge distillation methods in which the student model and the teacher model are the same one. It has attracted increasing attention considering the application scenario where the high-performance large teacher model is not available. In Zhang et al. (2019), the network are divided into multiple sections, and each section is deepened with extra layers as the teacher model. After that, there are several variants (Hou et al., 2019; Phuong & Lampert, 2019; Zhang & Sabuncu, 2020; Mobahi et al., 2020; Zheng et al., 2022) of self-distillation methods are proposed to achieve better performance.

# 3 METHOD

## 3.1 NETWORK DESIGN

**Backbone**  It has been shown that multi-branch networks (Szegedy et al., 2015; 2016; He et al., 2016; Huang et al., 2017) can often achieve better classification performance than single-path ones (Krizhevsky et al., 2012; Simonyan & Zisserman, 2014), but often it comes with the reduction of the parallelism and results in an increase of inference latency. On the contrary, plain single-path networks like VGG (Simonyan & Zisserman, 2014) take advantage of high parallelism and less memory footprint, leading to higher inference efficiency. Lately, in RepVGG (Ding et al., 2021), a structural re-parameterization method is proposed to decouple the training-time multi-branch topology with an

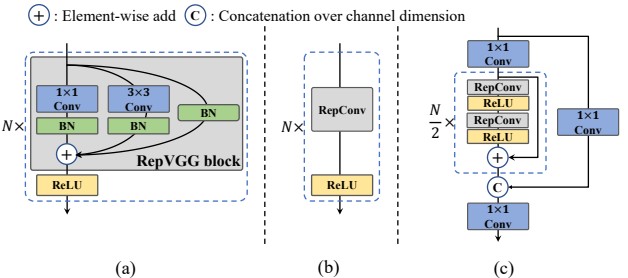

(a)          (b)          (c)

Figure 2: **(a)** RepBlock is composed of a stack of RepVGG blocks with ReLU activations at training. **(b)** During inference time, RepVGG block is converted to RepConv. **(c)** CSPStackRep Block comprises three 1×1 convolutional layers and a stack of sub-blocks of double RepConvs following the ReLU activations with a residual connection.

inference-time plain architecture to achieve a better speed-accuracy trade-off.

Inspired by the above, we design an efficient re-parameterizable backbone denoted as *EfficientRep*. For small models, the main component of the backbone is RepBlock during the training phase, as shown in Fig. 2 (a). And each RepBlock is converted to stacks of 3 × 3 convolutional layers (denoted as RepConv) with ReLU activation functions during the inference phase, as shown in Fig. 2 (b). Typically a 3×3 convolution is highly optimized on mainstream GPUs and CPUs and it enjoys

Figure 3: (a) The neck of YOLOv6 (N and S are shown). Note for M/L, RepBlocks is replaced with CSPStackRep. (b) The structure of a BiC module. (c) A SimCSPSPPF block.

higher computational density. Consequently, EfficientRep backbone sufficiently utilizes the computing power of the hardware, resulting in a significant decrease in inference latency while enhancing the representation ability in the meantime.

However, we notice that with the model capacity further expanded, the computation cost and the number of parameters in the single-path plain network grow exponentially. To achieve a better trade-off between the computation burden and accuracy, we revise a CSPStackRep Block to build the backbone of medium and large networks. As shown in Fig. 2 (c), CSPStackRep Block is composed of three $1 \times 1$ convolution layers and a stack of sub-blocks consisting of two RepVGG blocks (Ding et al., 2021) or RepConv (at training or inference respectively) with a residual connection. Besides, a *cross stage partial* (CSP) connection is adopted to boost performance without excessive computation cost. Compared with CSPRepResStage (Xu et al., 2022), it comes with a more succinct outlook and considers the balance between accuracy and speed.

**Neck** In practice, feature integration at multiple scales has been proven to be a critical and effective component of object detection. Feature Pyramid Network (FPN) (Lin et al., 2017a) is proposed to aggregate the high-level semantic features and low-level features via a top-down pathway, which provides more accurate localization. Subsequently, there have been several works (Liu et al., 2018; Tan et al., 2020; Ghiasi et al., 2019; Chen et al., 2021) on Bi-directional FPN to enhance the ability of hierarchical feature representation. PANet (Liu et al., 2018) adds an extra bottom-up pathway on top of FPN to shorten the information path of low-level and top-level features, which facilitates the propagation of accurate signals from low-level features. BiFPN (Tan et al., 2020) introduces learnable weights for different input features and simplifies PAN to achieve better performance with high efficiency. PRB-FPN (Chen et al., 2021) is proposed to retain high-quality features for accurate localization by a parallel FP structure with bi-directional fusion and associated improvements.

Motivated by the above, we adopt the modified PAN topology (Liu et al., 2018) from YOLOv4 (Bochkovskiy et al., 2020) and YOLOv5 (Glenn, 2022) as the base of our detection neck. We replace the CSP-Block used in YOLOv5 with RepBlock (for small models) or CSPStackRep Block (for large models) and adjust the width and depth accordingly. Besides, we design an enhanced PAN as our detection neck. To augment localization signals without bringing in excessive computation burden, we propose a Bi-directional Concatenation (BiC) module to integrate feature maps of three adjacent layers, which fuses an extra low-level feature from backbone $C_{i-1}$ into $P_i$ (Fig. 3). In this case, more accurate localization signals can be preserved, which is significant for the localization of small objects.

Moreover, we simplify the SPPF block (Glenn, 2022) to have a CSP-like version called SimCSP-SPPF Block, which strengthens the representational ability. Particularly, we revise the SimSPPC-SPC Block in (Wang et al., 2022) by shrinking the channels of hidden layers and retouching space pyramid pooling. In addition, we upgrade the CSPBlock with RepBlock (for small models) or CSPStackRep Block (for large models) and accordingly adjust the width and depth. The neck of YOLOv6 is denoted as RepBi-PAN, the framework of which is shown in Fig. 3.

**Lite models for mobile** To meet the needs of mobile scenarios, we introduce YOLOv6-Lite series. The backbones are a lightweight version of EfficientRep with scale factors (0.7, 1.1, 1.5 for S/M/L) and reduced channels. The neck selectively adopts the separable convolution (Howard et al., 2017),

and the head introduces the large-kernel (5×5) depthwise module. These models are designed to cater to high, medium, and low-end mobile chips. The results are later shown in Table 2.

## 3.2 ANCHOR-AIDED TRAINING

Detectors are made anchor-free to pursue a higher inference speed. However, we experimentally find that the anchor-based paradigm brings significant performance gains on YOLOv6-N under the same settings on small, medium and large objects, as shown in Table 1. In light of this, we conceive that the

Table 1: Anchor-based paradigm enjoys a great performance boost on YOLOv6-N.

| Paradigm | $AP^s$ | $AP^m$ | $AP^l$ |
|---|---|---|---|
| Anchor-free | 16.0% | 39.5% | 51.0% |
| Anchor-based | **17.2%** (**+1.2%**) | **39.8%** (**+0.3%**) | **52.1%** (**+1.1%**) |

anchor-based detector can learn from a different perspective from the anchor-free one in training to obtain better performance. However, the better speed performance of the anchor-free paradigm is of vital importance for real-time detectors. Therefore, we propose *anchor-aided training (AAT)*, in which the *anchor-based auxiliary branches* are introduced to combine the advantages of anchor-based and anchor-free paradigms. They are applied both in the classification and the regression head. Fig. 4 shows the detection head with the auxiliaries. During the training stage, the auxiliary branches and the anchor-free branches learn from independent losses while signals are propagated altogether. Therefore, additional embedded guidance information from auxiliary branches is integrated into the main anchor-free heads. Worth mentioning that the auxiliary branches are removed at inference, which boosts the accuracy without decreasing speed.

## 3.3 ADAPTIVE SELF-DISTILLATION

There are two sub-tasks in object detection: classification and localization. For classification, we apply the vanilla knowledge distillation technique by minimizing the KL-divergence between the class prediction of the teacher and the student. As for localization, our large models (i.e., YOLOv6-M/L) adopt DFL (Li et al., 2020) as regression loss for the convenience of performing self-distillation on localization (Zheng et al., 2022). The knowledge distillation loss is formulated as:

Figure 4: The detection head with anchor-based auxiliary branches during training. The auxiliary branches are removed at inference. 'af' and 'ab' are short for 'anchor-free' and 'anchor-based'.

$$L_{KD} = KL(p_t^{cls}||p_s^{cls}) + KL(p_t^{reg}||p_s^{reg}), \quad (1)$$

where $p_t^{cls}$ and $p_s^{cls}$ are class predictions of the teacher model and the student model respectively, and accordingly $p_t^{reg}$ and $p_s^{reg}$ are box regression predictions. The overall loss function is,

$$L_{total} = L_{det} + \alpha L_{KD}, \quad (2)$$

where $L_{det}$ is the detection loss computed with predictions and labels. The hyperparameter $\alpha$ is introduced to balance two losses. In the early stage of training, the soft labels from the teacher are easier to learn. As the training continues, the performance of the student will match the teacher so that the hard labels will help students more. We apply *cosine weight decay* to $\alpha$ to adaptively adjust the information from hard labels and soft ones from the teacher. The formulation of $\alpha$ is:

$$\alpha = -0.99 * ((1 - cos(\pi * E_i/E_{max}))/2) + 1, \quad (3)$$

where $E_i$ denotes the current training epoch and $E_{max}$ represents the maximum training epochs.

Notably, the introduction of DFL (Li et al., 2020) requires extra parameters for the regression branch, which affects the inference speed of small models significantly. Therefore, we specifically design the *Decoupled Localization Distillation* (DLD) for our small models (i.e., YOLOv6-N/S) to boost performance without speed degradation. Specifically, we append a heavy auxiliary *enhanced regression branch* to incorporate DFL.

Table 2: Comparisons with other YOLO-series and end-to-end detectors on COCO 2017 *val*.

| Method | Input Size | $AP^{val}$ | $AP_{50}^{val}$ | FPS (bs=1) | FPS (bs=32) | Latency (bs=1) | Params | FLOPs |
|---|---|---|---|---|---|---|---|---|
| YOLOv5-N | 640 | 28.0% | 45.7% | 602 | 735 | 1.7 ms | 1.9 M | 4.5 G |
| YOLOv5-S | 640 | 37.4% | 56.8% | 376 | 444 | 2.7 ms | 7.2 M | 16.5 G |
| YOLOv5-M | 640 | 45.4% | 64.1% | 182 | 209 | 5.5 ms | 21.2 M | 49.0 G |
| YOLOv5-L | 640 | 49.0% | 67.3% | 113 | 126 | 8.8 ms | 46.5 M | 109.1 G |
| YOLOv5-N6 | 1280 | 36.0% | 54.4% | 172 | 175 | 5.8 ms | 3.2 M | 18.4 G |
| YOLOv5-S6 | 1280 | 44.8% | 63.7% | 103 | 103 | 9.7 ms | 12.6 M | 67.2 G |
| YOLOv5-M6 | 1280 | 51.3% | 69.3% | 49 | 48 | 20.1 ms | 35.7 M | 200.0 G |
| YOLOv5-L6 | 1280 | 53.7% | 71.3% | 32 | 30 | 31.3 ms | 76.8 M | 445.6 G |
| YOLOv5-X6 | 1280 | 55.0% | 72.7% | 17 | 17 | 58.6 ms | 140.7 M | 839.2 G |
| YOLOX-Tiny | 416 | 32.8% | 50.3%* | 717 | 1143 | 1.4 ms | 5.1 M | 6.5 G |
| YOLOX-S | 640 | 40.5% | 59.3%* | 333 | 396 | 3.0 ms | 9.0 M | 26.8 G |
| YOLOX-M | 640 | 46.9% | 65.6%* | 155 | 179 | 6.4 ms | 25.3 M | 73.8 G |
| YOLOX-L | 640 | 49.7% | 68.0%* | 94 | 103 | 10.6 ms | 54.2 M | 155.6 G |
| PPYOLOE-S | 640 | 43.1% | 59.6% | 327 | 419 | 3.1 ms | 7.9 M | 17.4 G |
| PPYOLOE-M | 640 | 49.0% | 65.9% | 152 | 189 | 6.6 ms | 23.4 M | 49.9 G |
| PPYOLOE-L | 640 | 51.4% | 68.6% | 101 | 127 | 10.1 ms | 52.2 M | 110.1 G |
| YOLOv7-Tiny | 416 | 33.3%* | 49.9%* | 787 | 1196 | 1.3 ms | 6.2 M | 5.8 G |
| YOLOv7-Tiny | 640 | 37.4%* | 55.2%* | 424 | 519 | 2.4 ms | 6.2 M | 13.7 G* |
| YOLOv7 | 640 | 51.2% | 69.7%* | 110 | 122 | 9.0 ms | 36.9 M | 104.7 G |
| YOLOv7-E6E | 1280 | 56.8% | 74.4%* | 16 | 17 | 59.6 ms | 151.7 M | 843.2 G |
| YOLOv8-N | 640 | 37.3% | 52.6%* | 561 | 734 | 1.8 ms | 3.2 M | 8.7 G |
| YOLOv8-S | 640 | 44.9% | 61.8%* | 311 | 387 | 3.2 ms | 11.2 M | 28.6 G |
| YOLOv8-M | 640 | 50.2% | 67.2%* | 143 | 176 | 7.0 ms | 25.9 M | 78.9 G |
| YOLOv8-L | 640 | 52.9% | 69.8%* | 91 | 105 | 11.0 ms | 43.7 M | 165.2 G |
| DETR-DC5 (R50) | 800×1333 | 43.3% | 63.1% | - | - | - | 41 M | 187 G |
| DETR-DC5 (R101) | 800×1333 | 44.9% | 64.7% | - | - | - | 60 M | 253 G |
| Deformable-DETR | 800×1333 | 46.2% | 65.2% | - | - | - | 47 M | 279 G |
| YOLOv6Lite-S | 320×320 | 22.4% | 34.3% | - | - | 7.99 ms† | 0.55 M | 0.56 G |
| YOLOv6Lite-M | 320×320 | 25.1% | 38.1% | - | - | 9.08 ms† | 0.79 M | 0.67 G |
| YOLOv6Lite-L | 320×320 | 28.0% | 41.9% | - | - | 11.37 ms† | 1.09 M | 0.87 G |
| YOLOv6-N | 640 | 37.0% / 37.5%‡ | 52.7% / 53.1%‡ | 779 | 1187 | 1.3 ms | 4.7 M | 11.4 G |
| YOLOv6-S | 640 | 44.3% / 45.0%‡ | 61.2% / 61.8%‡ | 339 | 484 | 2.9 ms | 18.5 M | 45.3 G |
| YOLOv6-M | 640 | 49.1% / 50.0%‡ | 66.1% / 66.9%‡ | 175 | 226 | 5.7 ms | 34.9 M | 85.8 G |
| YOLOv6-L | 640 | 51.8% / 52.8%‡ | 69.2% / 70.3%‡ | 98 | 116 | 10.3 ms | 59.6 M | 150.7 G |
| YOLOv6-N6 | 1280 | 44.9% | 61.5% | 228 | 281 | 4.4 ms | 10.4 M | 49.8 G |
| YOLOv6-S6 | 1280 | 50.3% | 67.7% | 98 | 108 | 10.2 ms | 41.4 M | 198.0 G |
| YOLOv6-M6 | 1280 | 55.2%‡ | 72.4%‡ | 47 | 55 | 21.0 ms | 79.6 M | 379.5 G |
| YOLOv6-L6 | 1280 | 57.2%‡ | 74.5%‡ | 26 | 29 | 38.5 ms | 140.4 M | 673.4 G |

[a] FPS and latency are measured in TensorRT FP16 on an NVIDIA Tesla T4 GPU. Both the accuracy and the speed are evaluated with the input resolution of 640×640. Exception for YOLOv6Lite models†, which are tested on Qualcomm 888 (sm8350) mobile chip with MNN with 2 threads.
[b] '‡' represents that the proposed self-distillation method is utilized.
[c] '*' represents the re-evaluated result of the released model through the official code.
[d] The latency and throughput of DETRs are not tested being non real-time.

During the self-distillation, the student is equipped with a naïve regression branch and the enhanced regression branch while the teacher only uses the auxiliary branch. Note that the naïve regression branch is only trained with hard labels while the auxiliary is updated according to signals from both the teacher and hard labels. After the distillation, the naïve regression branch is retained whilst the auxiliary branch is removed. With this strategy, the advantages of the heavy regression branch for DFL in distillation are considerably maintained without impacting the inference efficiency.

# 4 EXPERIMENTS

## 4.1 COMPARISON OF REAL-TIME DETECTORS

We compare with the state-of-the-art methods and the results are shown in Table 2 and Fig. 1. The evaluation is focused on the throughput and the GPU latency at deployment. Generally, YOLOv6 comes with the best speed performance in terms of both throughput and latency. For detecting extra-large objects, we follow (Glenn, 2022) to add an extra stage on the top of the backbone to have a feature (C6) at a higher level, and the neck is expanded accordingly. Further, the image resolution is adapted from 640 to 1280. The feature strides range from 8 to 64, which benefits the accurate detection of rather small and extra-large objects in high-resolution images. The YOLOv6 of all sizes with C6 features are named YOLOv6-N6/S6/M6/L6 respectively, which obtain significant gains in accuracy. We also present our lightweight versions made for mobile devices. Also to note that Transformer-based end-to-end detectors like DETR (Carion et al., 2020), Deformable-DETR (Zhu et al., 2020) are less competitive compared to YOLO models at similar FLOPs.

## 4.2 DOWNSTREAM TASKS

**Face Keypoint** We extend YOLOv6 to a face detector with landmarks supervision. The key modifications are summarized as follows. Firstly, we add a regression head for 5-keypoint landmarks based on object regression branch. With the extra supervision, it makes the face detector more accurate. Secondly, we modify the efficient decoupled head with the same-channel strategy instead of hybrid-channel, which increases the capability to detect tiny faces. Considering dense scenes, the repulsion loss is

Table 3: Comparison of YOLOv6-face and YOLO5-face on WIDER FACE dataset. *: Originally reported.

| Model | Size | Easy | Medium | Hard | FPS (bs=1) | FPS (bs=32) | Params (M) | FLOPs (G) |
|-------|------|------|--------|------|------|------|------|------|
| MTCNN | - | 85.1 | 82.9 | 76.1 | 99* | - | - | - |
| YOLOv5-face-S | 640 | 94.3 | 92.6 | 83.2 | 464 | 622 | 7.06 | 15.2 |
| YOLOv5-face-M | 640 | 95.3 | 93.8 | 85.3 | 217 | 262 | 21.04 | 48.2 |
| YOLOv5-face-L | 640 | 95.9 | 94.4 | 84.5 | 132 | 149 | 46.6 | 110.6 |
| YOLOv6-face-S | 640 | 96.2 | 94.7 | 85.1 | 339 | 484 | 12.41 | 32.45 |
| YOLOv6-face-M | 640 | 97 | 95.3 | 86.3 | 188 | 240 | 24.85 | 70.59 |
| YOLOv6-face-L | 640 | 97.2 | 95.9 | 87.5 | 102 | 121 | 56.77 | 159.24 |

applied to improve detecting performance and reduce the sensitivity of NMS thresholds. We implement a series of face detector models at S/M/L scales, and compare the performance with YOLO5face (Qi et al., 2022) on WIDER FACE (Yang et al., 2016) validation set in table 3. We see that YOLOv6-face outperforms YOLO5-face and MTCNN (Zhang et al., 2017b) on the easy, medium, and hard subsets.

**Instance Segmentation** Based on YOLOv6, we add a mask branch, paralleling to the regression branch, which generates prototype masks, following YOLACT (Bolya et al., 2019). Besides, mask coefficients are predicted on the regression branch by adding extra two convolution kernels after multi-level features. By linearly multiplying the prototype masks with mask coefficients, we can obtain the final instance segmentation results. In Table 4, we compare YOLOv6 n/s/m/l/x segmentation models on the COCO validation dataset with other real-time instance like YOLOv8 and RTMDet-Ins (Lyu et al., 2022).

YOLOv6-seg models at full scales obtain higher mask AP than YOLOv8-seg

Table 4: Comparison of YOLOv6-seg with previous methods on COCO val2017. *: Originally reported.

| Model | Size | mAP$^{box}$ | mAP$^{mask}$ | FPS (bs=1) | Params (M) | FLOPs (G) |
|-------|------|-------------|--------------|------|------|------|
| YOLACT | 550 | - | 29.8 | 33.5* | - | - |
| YOLOv8-seg-N | 640 | 36.7 | 30.5 | 521 | 3.4 | 12.6 |
| YOLOv8-seg-S | 640 | 44.6 | 36.8 | 286 | 11.8 | 42.6 |
| YOLOv8-seg-M | 640 | 49.9 | 40.8 | 137 | 27.3 | 110.2 |
| YOLOv8-seg-L | 640 | 52.3 | 42.6 | 88 | 46 | 220.5 |
| YOLOv8-seg-X | 640 | 53.4 | 43.4 | 56 | 71.8 | 344.1 |
| RTMDet-Ins-S | 640 | 44 | 38.7 | 243 | 10.2 | 21.5 |
| RTMDet-Ins-M | 640 | 48.8 | 42.1 | 116 | 27.6 | 54.1 |
| RTMDet-Ins-L | 640 | 51.2 | 43.7 | 70 | 57.4 | 106.6 |
| RTMDet-Ins-X | 640 | 52.4 | 44.6 | 40 | 102.7 | 182.7 |
| YOLOv6-seg-N | 640 | 35.3 | 31.2 | 645 | 4.9 | 14 |
| YOLOv6-seg-S | 640 | 44 | 38 | 292 | 19.6 | 55.5 |
| YOLOv6-seg-M | 640 | 48.2 | 41.3 | 148 | 37.1 | 108.5 |
| YOLOv6-seg-L | 640 | 51.1 | 43.7 | 93 | 63.6 | 191 |
| YOLOv6-seg-X | 640 | 52.2 | 44.8 | 47 | 119.1 | 351 |

models with similar inference speed. Specifically, YOLOv6x-seg achieves 44.8% mask AP, surpassing the previous best practice RTMDet-Ins-x by 0.2% AP with 17.5% faster speed.

## 4.3 ABLATION STUDY

### 4.3.1 NETWORK DESIGN

**RepBlock and CSPStackRep Block** We compare the single-path structure and multi-branch structure on backbones and necks, as well as the channel coefficient (denoted as CC) of CSPStackRep Block. All models described in this part adopt TAL as the label assignment strategy, VFL as the classification loss, and GIoU with DFL as the regression loss. Results are shown in Table 5. We find that the optimal network structure for models with different sizes should come up with different solutions.

Table 5: Ablation study on backbones and necks.

| Models | Block | CC | $AP^{val}$ | FPS (bs=32) | Params | FLOPs |
|---|---|---|---|---|---|---|
| YOLOv6-N | RepBlock | - | 35.2% | 1237 | 4.3M | 11.1G |
| | CSPStackRep Block | 1/2 | 32.7% | 1257 | 2.3M | 5.6G |
| YOLOv6-S | RepBlock | - | 43.2% | 499 | 17.2M | 44.2G |
| | CSPStackRep Block | 1/2 | 43.4% | 511 | 11.5M | 27.7G |
| YOLOv6-M | RepBlock | - | 47.9% | 137 | 67.1M | 175.6G |
| | CSPStackRep Block | 2/3 | 48.1% | 237 | 34.3M | 82.2G |
| | CSPStackRep Block | 1/2 | 47.3% | 237 | 27.7M | 68.4G |
| YOLOv6-L | CSPStackRep Block | 2/3 | 50.1% | 149 | 54.7M | 142.7G |
| | CSPStackRep Block | 1/2 | 50.1% | 151 | 58.5M | 144.0G |

For YOLOv6-N, the single-path structure outperforms the multi-branch structure in terms of both accuracy and speed. Although the single-path structure has more FLOPs and parameters than the multi-branch structure, it could run faster due to a relatively lower memory footprint and a higher degree of parallelism. For YOLOv6-S, the two block styles bring similar performance. When it comes to larger models, multi-branch structure achieves better performance in accuracy and speed. We select the multi-branch with a channel coefficient of 2/3 for YOLOv6-M and 1/2 for YOLOv6-L.

**BiC** We conducted a series of experiments to verify the effectiveness of the proposed BiC module. As can be seen in Table 6, applying the BiC module only on the top-down pathway of PAN brings 0.6%/0.4% AP improvements on YOLOv6-S/L respectively with negligible loss of efficiency. In contrast, when we try to import the BiC module into the bottom-up pathway, no positive gain in accuracy is obtained. The probable reason is that the BiC module on the bottom-up pathway would lead to confusion for detection heads about features at different scales. Therefore, we merely adopt the BiC module on the top-down pathway. Besides, the results indicate that the BiC module gives an impressive boost to the performance of small object detection. For both YOLOv6-S and YOLOv6-L, the detection performance on small objects is improved by 1.8%.

Table 6: Effectiveness of the BiC module on YOLOv6 models. BU: Bottom-up, TD: Top-down

| Model | BiC BU | BiC TD | $AP^{val}$ | $AP^s$ | $AP^m$ | $AP^l$ | FPS (bs=32) |
|---|---|---|---|---|---|---|---|
| YOLOv6-S | | | 43.1% | 23.4% | 48.0% | 59.9% | **513** |
| | | ✓ | **43.7%** | **25.2%** | **48.7%** | **60.4%** | 492 |
| | ✓ | ✓ | **43.7%** | 25.0% | **48.7%** | 59.7% | 485 |
| YOLOv6-L | | | 50.9% | 32.4% | 56.0% | **68.0%** | **125** |
| | | ✓ | **51.3%** | **34.2%** | 56.5% | 67.6% | 120 |
| | ✓ | ✓ | 51.1% | 33.6% | **56.7%** | 67.9% | 119 |

**SimCSPSPPF** Further, we explore the influence of different types of SPP Blocks, including the simplified variants of SPPF (Glenn, 2022) and SPPCSPC (Wang et al., 2022) (denoted as SimSPPF and SimSPPCSPC respectively) and our SimCSPSPPF blocks.

Additionally, we apply SimSPPF blocks on the top three feature maps (P3, P4, and P5) of our backbone to verify its effectiveness, which is denoted as SimSPPF*3. Experimental results are shown in Table 7. We observe that heavily adopting SimSPPF brings little gain in accuracy with the increased computational complexity. SimSPPCSPC outperforms SimSPPF by 1.6%/0.3% AP on YOLOv6-N/S respectively while significantly decreasing inference speed. Compared with SimSPPF, our SimCSPSPPF version can obtain

Table 7: Ablation study on SPP Blocks.

| Model | SPP Blocks | $AP^{val}$ | FPS (bs=32) |
|---|---|---|---|
| YOLOv6-N | SimSPPF | 35.8% | **1190** |
| | SimSPPF*3 | 35.9% | 1072 |
| | SimSPPCSPC | **37.4%** | 1078 |
| | SimCSPSPPF | 36.9% | 1176 |
| YOLOv6-S | SimSPPF | 43.7% | **492** |
| | SimSPPF*3 | 43.6% | 447 |
| | SimSPPCSPC | 44.0% | 432 |
| | SimCSPSPPF | **44.1%** | 477 |
| YOLOv6-M | SimSPPF | 48.6% | **227** |
| | SimCSPSPPF | **48.7%** | 218 |
| YOLOv6-L | SimSPPF | **51.3%** | **120** |
| | SimCSPSPPF | 51.1% | 117 |

1.1%/0.4%/0.1% performance gain for YOLOv6-N/S/M respectively. In terms of inference efficiency, our SimCSPSPPF block runs nearly 10% faster than SimSPPCSPC and is slightly slower than SimSPPF. For a better accuracy-efficiency trade-off, the SimCSPSPPF blocks are introduced in YOLOv6-N/S. For YOLOv6-M/L, SimSPPF blocks are adopted.

### 4.3.2 ANCHOR-AIDED TRAINING

The advantages of the AAT are verified in YOLOv6. As shown in Table 8, it brings about 0.3%/0.5%/0.5% AP gain for YOLOv6-S/M/L respectively. Notably, the accuracy performance on small objects ($AP^s$) is significantly enhanced for YOLOv6-N/S/M. For YOLOv6-L, the performance on large objects ($AP^l$) is improved even further.

Table 8: Ablation on Anchor-aided Training.

| Method | AAT | $AP^{val}$ | $AP^s$ | $AP^m$ | $AP^l$ |
|---|---|---|---|---|---|
| YOLOv6-N | | **36.9%** | 17.2% | 41.1% | 52.9% |
| | ✓ | **36.9%** | **18.7%** | **41.2%** | **53.0%** |
| YOLOv6-S | | 44.1% | 24.7% | 48.7% | **61.1%** |
| | ✓ | **44.4%** | **25.4%** | **49.6%** | 60.2% |
| YOLOv6-M | | 48.6% | 29.7% | 53.7% | **65.5%** |
| | ✓ | **49.1%** | **31.1%** | **54.0%** | 65.4% |
| YOLOv6-L | | 51.3% | **34.2%** | 56.5% | 67.6% |
| | ✓ | **51.8%** | 33.4% | **56.8%** | **68.8%** |

### 4.3.3 ADAPTIVE SELF-DISTILLATION

We firstly verify the proposed adaptive self-distillation method on YOLOv6-L. For a fair comparison, we also verified the model performance by doubling the training epochs besides the baseline since the self-distillation needs an extra training cycle to obtain the teacher model. As seen in Table 9, no performance improvement is attained without the weight decay strategy compared with the baseline. Doubling the training epochs without self-distillation is even worse due to overfitting. After the introduction of weight decay, the model is boosted by 0.6% AP.

In addition, the DLD specifically designed for small models is ablated on YOLOv6-S. As per self-distillation for large models, we also compare the results with the model trained with doubled epochs. As shown in Table 10, YOLOv6-S with DLD gives 0.7% AP boost and performs 0.5% better than that of training with doubled epochs.

These experiment results on both small and large models demonstrate that our self-distillation strategy is more cost-effective than training for longer epochs. Please refer Appendix B for more conclusions about other design and technique choices.

Table 9: Ablation study on the self-distillation on YOLOv6-L.

| Model | Weight Decay | $AP^{val}$ |
|---|---|---|
| Baseline | ✗ | 51.8% |
| Double epochs | ✗ | 51.7% |
| Self-distillation | ✗ | 51.8% |
| | ✓ | **52.4%** |

Table 10: Ablation study of DLD on YOLOv6-S.

| DLD | Double epochs | $AP^{val}$ |
|---|---|---|
| ✗ | ✗ | 44.4% |
| ✗ | ✓ | 44.6% |
| ✓ | ✗ | **45.1%** |

## 5 CONCLUSION

In a nutshell, YOLOv6 is built with a myriad of brand-new architectural compositions (namely RepBlock, CSPStackRep, RepBi-PAN, BiC, and SimCSPSPPF) and a suite of novel paralleled training schemes like anchor-aided training, adaptive self-distillation via a removable auxiliary regression branch, along with other carefully chosen techniques, altogether bringing it to a next level of cutting-edge performance and speed. Downstream tasks like facial keypoint detection and instance segmentation are easily implemented and enjoy significant improvement. We believe YOLOv6 at all scales is a competitive real-time object detector and it will greatly facilitate real-world applications on mobile, IoT, and high-end cloud computing devices.

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

# A  ADDITIONAL DESIGNS AND CHOICES

## A.1  NETWORK

**Efficient decoupled head**  The detection head of YOLOv5 is a coupled head with parameters shared between the classification and localization branches, while its counterparts in FCOS (Tian et al., 2019) and YOLOX (Ge et al., 2021b) decouple the two branches, and additional two 3×3 convolutional layers are introduced in each branch to boost the performance.

In YOLOv6, we adopt a *hybrid-channel* strategy to build a more efficient decoupled head. Specifically, we reduce the number of the middle 3×3 convolutional layers to only one. The width of the head is jointly scaled by the width multiplier for the backbone and the neck. These modifications further reduce computation costs to achieve a lower inference latency.

**Anchor-free**  Anchor-free detectors stand out because of their better generalization ability and simplicity in decoding prediction results. The time cost of its post-processing is substantially reduced. There are two types of anchor-free detectors: anchor point-based (Tian et al., 2019; Ge et al., 2021b) and keypoint-based (Zhou et al., 2019; Law & Deng, 2018; Yang et al., 2019). In YOLOv6, we adopt the anchor point-based paradigm, whose box regression branch actually predicts the distance from the anchor point to the four sides of the bounding boxes.

## A.2  LABEL ASSIGNMENT

Label assignment is responsible for assigning labels to predefined anchors during the training stage. Previous work has proposed various label assignment strategies ranging from simple IoU-based strategy and inside ground-truth method (Tian et al., 2019) to other more complex schemes (Zhang et al., 2020b; Ge et al., 2021b; Feng et al., 2021; Li et al., 2022; Zand et al., 2022).

**SimOTA**  OTA (Ge et al., 2021a) considers the label assignment in object detection as an optimal transmission problem. It defines positive/negative training samples for each ground-truth object from a global perspective. SimOTA (Ge et al., 2021b) is a simplified version of OTA (Ge et al., 2021a), which reduces additional hyperparameters and maintains the performance. SimOTA was utilized as the label assignment method in the early version of YOLOv6. However, in practice, we find that introducing SimOTA will slow down the training process. And it is not rare to fall into unstable training. Therefore, we desire a replacement for SimOTA.

**Task alignment learning**  Task Alignment Learning (TAL) was first proposed in TOOD (Feng et al., 2021), in which a unified metric of classification score and predicted box quality is designed. The IoU is replaced by this metric to assign object labels. To a certain extent, the problem of the misalignment of tasks (classification and box regression) is alleviated.

The other main contribution of TOOD is about the task-aligned head (T-head). T-head stacks convolutional layers to build interactive features, on top of which the Task-Aligned Predictor (TAP) is used. PP-YOLOE (Xu et al., 2022) improved T-head by replacing the layer attention in T-head with the lightweight ESE attention, forming ET-head. However, we find that the ET-head will deteriorate the inference speed in our models and it comes with no accuracy gain. Therefore, we retain the design of our Efficient decoupled head. Furthermore, we observed that TAL could bring more performance improvement than SimOTA and stabilize the training. Therefore, we adopt TAL as our default label assignment strategy in YOLOv6.

## A.3  LOSS FUNCTIONS

Object detection contains two sub-tasks: *classification* and *localization*, corresponding to two loss functions: *classification loss* and *box regression loss*. For each sub-task, there are various loss functions presented in recent years. In this section, we will introduce these loss functions and describe how we select the best ones for YOLOv6.

### A.3.1 CLASSIFICATION LOSS

Improving the performance of the classifier is a crucial part of optimizing detectors. Focal Loss (Lin et al., 2017b) modified the traditional cross-entropy loss to solve the problems of class imbalance either between positive and negative examples, or hard and easy samples. To tackle the inconsistent usage of the quality estimation and classification between training and inference, Quality Focal Loss (QFL) (Li et al., 2020) further extended Focal Loss with a joint representation of the classification score and the localization quality for the supervision in classification. Whereas VariFocal Loss (VFL) (Zhang et al., 2021) is rooted from Focal Loss (Lin et al., 2017b), but it treats the positive and negative samples asymmetrically. By considering positive and negative samples at different degrees of importance, it balances learning signals from both samples. Poly Loss (Leng et al., 2022) decomposes the commonly used classification loss into a series of weighted polynomial bases. It tunes polynomial coefficients on different tasks and datasets, which is proved better than Cross-entropy Loss and Focal Loss through experiments. We assess all these advanced classification losses on YOLOv6 to finally adopt VFL (Zhang et al., 2021).

### A.3.2 LOCALIZATION LOSS

**Box Regression Loss**   Box regression loss provides significant learning signals localizing bounding boxes precisely. L1 loss is the original box regression loss in early works. Progressively, a variety of well-designed box regression losses have sprung up, such as IoU-series loss (Yu et al., 2016; Zheng et al., 2020; Rezatofighi et al., 2019; Zheng et al., 2020; He et al., 2021; Gevorgyan, 2022) and probability loss (Li et al., 2020).

**IoU-series Loss**   IoU loss (Yu et al., 2016) regresses the four bounds of a predicted box as a whole unit. It has been proved to be effective because of its consistency with the evaluation metric. There are many variants of IoU, such as GIoU (Rezatofighi et al., 2019), DIoU (Zheng et al., 2020), CIoU (Zheng et al., 2020), $\alpha$-IoU (He et al., 2021) and SIoU (Gevorgyan, 2022), etc, forming relevant loss functions. We experiment with GIoU, CIoU and SIoU in this work. And SIoU is applied to YOLOv6-N and YOLOv6-T, while others use GIoU.

**Probability Loss**   Distribution Focal Loss (DFL) (Li et al., 2020) simplifies the underlying continuous distribution of box locations as a discretized probability distribution. It considers ambiguity and uncertainty in data without introducing any other strong priors, which is helpful to improve the box localization accuracy especially when the boundaries of the ground-truth boxes are blurred. Upon DFL, DFLv2 (Li et al., 2021) develops a lightweight sub-network to leverage the close correlation between distribution statistics and the real localization quality, which further boosts the detection performance. However, DFL usually outputs $17\times$ more regression values than general box regression, leading to a substantial overhead. The extra computation cost significantly hinders the training of small models. Whilst DFLv2 further increases the computation burden because of the extra sub-network. In our experiments, DFLv2 brings similar performance gain to DFL on our models. Consequently, we only adopt DFL in YOLOv6-M/L. Experimental details can be found in Appendix B.3.

### A.3.3 OBJECT LOSS

Object loss was first proposed in FCOS (Tian et al., 2019) to reduce the score of low-quality bounding boxes so that they can be filtered out in post-processing. It was also used in YOLOX (Ge et al., 2021b) to accelerate convergence and improve network accuracy. As an anchor-free framework like FCOS and YOLOX, we have tried object loss into YOLOv6. Unfortunately, it doesn't bring many positive effects. Details are given in Section 4.

## B   DETAILED EXPERIMENTAL RESULTS

We demonstrate the detail ablation experimental results about the components of basic architecture of YOLOv6 in this section.

## B.1 Ablation on Network Design

**Combinations of convolutional layers and activation functions**   YOLO series adopted a wide range of activation functions, ReLU Nair & Hinton (2010), LReLU Maas et al. (2013), Swish Ramachandran et al. (2017), SiLU Elfwing et al. (2018), Mish Misra (2019) and so on. Among these activation functions, SiLU is the most used. Generally speaking, SiLU performs with better accuracy and does not cause too much extra computation cost. However, when it comes to industrial applications, especially for deploying models with TensorRT NVIDIA (2018) acceleration, ReLU has a greater speed advantage because of its fusion into convolution. Moreover, we further verify the effectiveness of combinations of RepConv/ordinary convolution (denoted as Conv) and ReLU/SiLU/LReLU in networks of different sizes to achieve a better trade-off. As shown in Table 12, Conv with SiLU performs the best in accuracy while the combination of RepConv and ReLU achieves a better trade-off. We suggest users adopt RepConv with ReLU in latency-sensitive applications. We choose to use RepConv/ReLU combination in YOLOv6-N/T/S/M for higher inference speed and use the Conv/SiLU combination in the large model YOLOv6-L to speed up training and improve performance.

| Width | Depth | $AP^{val}$ | FPS (bs=32) | Params | FLOPs |
|---|---|---|---|---|---|
| [192, 384, 768] | 5 | 50.8% | 123 | 58.6 M | 144.7 G |
| [128, 256, 512] | 12 | 51.0% | 122 | 58.5 M | 144.0 G |

Table 11: Ablation study on the neck settings of YOLOv6-L. SiLU is selected as the activation function.

**Miscellaneous design**   We also conduct a series of ablation on other network parts mentioned in Section 3.1 based on YOLOv6-N. We choose YOLOv5-N as the baseline and add other components incrementally. Results are shown in Table 13. Firstly, with decoupled head (denoted as DH), our model is 1.4% more accurate with 5% increase in time cost. Secondly, we verify that the anchor-free paradigm is 51% faster than the anchor-based one for its $3\times$ less predefined anchors, which results in less dimensionality of the output. Further, the unified modification of the backbone (EfficientRep Backbone) and the neck (Rep-PAN neck), denoted as EB+RN, brings 3.6% AP improvements, and runs 21% faster. Finally, the optimized decoupled head (hybrid channels, HC) brings 0.2% AP and 6.8% FPS improvements in accuracy and speed respectively.

## B.2 Ablation on Label Assignment

In Table 14, we analyze the effectiveness of mainstream label assign strategies. Experiments are conducted on YOLOv6-N. As expected, we observe that SimOTA and TAL are the best two strate-

| Model | Conv. | Act. | $AP^{val}$ | FPS (bs=32) |
|---|---|---|---|---|
| YOLOv6-N | Conv | SiLU | **36.6%** | 963 |
| | RepConv | SiLU | 36.5% | 971 |
| | Conv | ReLU | 34.8% | **1246** |
| | RepConv | ReLU | 35.2% | 1233 |
| | Conv | LReLU | 35.4% | 983 |
| | RepConv | LReLU | 35.6% | 975 |
| YOLOv6-M | Conv | SiLU | **48.9%** | 180 |
| | RepConv | SiLU | **48.9%** | 180 |
| | Conv | ReLU | 47.7% | 235 |
| | RepConv | ReLU | 48.1% | **236** |
| | Conv | LReLU | 48.0% | 185 |
| | RepConv | LReLU | 48.1% | 187 |

Table 12: Ablation study on combinations of different types of convolutional layers (denoted as Conv.) and activation layers (denoted as Act.).

| DH | AF | EB+RN | HC | $\mathbf{AP}^{val}$ | FPS (bs=32) |
|----|----|-------|----|----|----|
| ✗ | ✗ | ✗ | ✗ | 28.0% | 672 |
| ✓ | ✗ | ✗ | ✗ | 29.4% | 637 |
| ✓ | ✓ | ✗ | ✗ | 30.7% | 962 |
| ✓ | ✓ | ✓ | ✗ | 34.3% | 1163 |
| ✓ | ✓ | ✓ | ✓ | **34.5%** | **1242** |

Table 13: Ablation study on all network designs in an incremental way. FPS is tested with FP16-precision and batch-size=32 on Tesla T4 GPUs.

| Method | $\mathbf{AP}^{val}$ |
|--------|----|
| ATSS Zhang et al. (2020b) | 32.5% |
| SimOTA Ge et al. (2021b) | 34.5% |
| TAL Feng et al. (2021) | **35.0%** |
| DW Li et al. (2022) | 33.4% |
| ObjectBox Zand et al. (2022) | 30.1% |

Table 14: Comparisons of label assignment methods.

gies. Compared with the ATSS, SimOTA can increase AP by 2.0%, and TAL brings 0.5% higher AP than SimOTA. Considering the stable training and better accuracy performance of TAL, we adopt TAL as our label assignment strategy.

In addition, the implementation of TOOD Feng et al. (2021) adopts ATSS Zhang et al. (2020b) as the warm-up label assignment strategy during the early training epochs. We also retain the warm-up strategy and further make some explorations on it. Details are shown in Table 15, and we can find that without warm-up or warmed up by other strategies (i.e., SimOTA) it can also achieve the similar performance.

## B.3 ABLATION ON LOSS FUNCTIONS

In the object detection framework, the loss function is composed of a classification loss, a box regression loss and an optional object loss, which can be formulated as follows:

$$L_{det} = L_{cls} + \lambda L_{reg} + \mu L_{obj}, \tag{4}$$

where $L_{cls}$, $L_{reg}$ and $L_{obj}$ are classification loss, regression loss and object loss. $\lambda$ and $\mu$ are hyperparameters.

In this subsection, we evaluate each loss function on YOLOv6. Unless otherwise specified, the baselines for YOLOv6-N, YOLOv6-S and YOLOv6-M are 35.0%, 42.9% and 48.0% trained with TAL, Focal Loss and GIoU Loss.

**Classification Loss**  We experiment Focal Loss Lin et al. (2017b), Poly lossLeng et al. (2022), QFL Li et al. (2020) and VFL Zhang et al. (2021) on YOLOv6-N/S/M. As can be seen in Table 16, VFL brings 0.2%/0.3%/0.1% AP improvements on YOLOv6-N/S/M respectively compared with Focal Loss. We choose VFL as the classification loss function.

| Warmup strategy | $\mathbf{AP}^{val}$ |
|-----------------|----|
| w/o | 34.9% |
| ATSS Zhang et al. (2020b) | **35.0%** |
| SimOTA Ge et al. (2021b) | 34.9% |

Table 15: Comparisons of label assignment methods in warm-up stage.

| Model | Classification Loss | $AP^{val}$ |
|---|---|---|
| YOLOv6-N | Focal Loss Lin et al. (2017b) | 35.0% |
| | Poly Loss Leng et al. (2022) | 34.0% |
| | QFL Li et al. (2020) | **35.4%** |
| | VFL Zhang et al. (2021) | 35.2% |
| YOLOv6-S | Focal Loss Lin et al. (2017b) | 42.9% |
| | Poly Loss Leng et al. (2022) | 41.5% |
| | QFL Li et al. (2020) | 43.1% |
| | VFL Zhang et al. (2021) | **43.2%** |
| YOLOv6-M | Focal Loss Lin et al. (2017b) | 48.0% |
| | Poly Loss Leng et al. (2022) | 46.9% |
| | QFL Li et al. (2020) | 48.0% |
| | VFL Zhang et al. (2021) | **48.1%** |

Table 16: Ablation study on classification loss functions.

**Regression Loss** IoU-series and probability loss functions are both experimented with on YOLOv6-N/S/M.

The latest IoU-series losses are utilized in YOLOv6-N/S/M. Experiment results in Table 17 show that SIoU Loss outperforms others for YOLOv6-N and YOLOv6-T, while CIoU Loss performs better on YOLOv6-M.

For probability losses, as listed in Table 18, introducing DFL can obtain 0.2%/0.1%/0.2% performance gain for YOLOv6-N/S/M respectively. However, the inference speed is greatly affected for small models. Therefore, DFL is only introduced in YOLOv6-M/L.

| Model | Loss | $AP^{val}$ |
|---|---|---|
| YOLOv6-N | GIoU Rezatofighi et al. (2019) | 35.1% |
| | CIoU Zheng et al. (2020) | 35.1% |
| | SIoU Gevorgyan (2022) | **35.5%** |
| YOLOv6-S | GIoU Rezatofighi et al. (2019) | 43.1% |
| | CIoU Zheng et al. (2020) | 43.1% |
| | SIoU Gevorgyan (2022) | **43.3%** |
| YOLOv6-M | GIoU Rezatofighi et al. (2019) | 48.2% |
| | CIoU Zheng et al. (2020) | **48.3%** |
| | SIoU Gevorgyan (2022) | 48.1% |

Table 17: Ablation study on IoU-series box regression loss functions. The classification loss is VFL Zhang et al. (2021).

| Method | Loss | $AP^{val}$ | FPS (bs=32) |
|---|---|---|---|
| YOLOv6-N | w/o | 35.0% | **1226** |
| | DFL Li et al. (2020) | **35.2%** | 1022 |
| | DFLv2 Li et al. (2021) | **35.2%** | 819 |
| YOLOv6-S | w/o | 42.9% | **486** |
| | DFL Li et al. (2020) | **43.0%** | 461 |
| | DFLv2 Li et al. (2021) | **43.0%** | 422 |
| YOLOv6-M | w/o | 48.0% | 233 |
| | DFL Li et al. (2020) | 48.2% | **236** |
| | DFLv2 Li et al. (2021) | **48.3%** | 226 |

Table 18: Ablation study on probability loss functions.

**Object Loss** Object loss is also experimented with YOLOv6, as shown in Table 19. From Table 19, we can see that object loss has negative effects on YOLOv6-N/S/M networks, where the

| Method | Object Loss | $\mathbf{AP}^{val}$ |
|--------|:-----------:|-----|
| YOLOv6-N | ✗ | **35.0**% |
|          | ✓ | 33.9% |
| YOLOv6-S | ✗ | **42.9**% |
|          | ✓ | 41.4% |
| YOLOv6-M | ✗ | **48.0**% |
|          | ✓ | 46.5% |

Table 19: Effectiveness of object loss.

maximum decrease is 1.1% AP on YOLOv6-N. The negative gain may come from the conflict between the object branch and the other two branches in TAL. Specifically, in the training stage, IoU between predicted boxes and ground-truth ones, as well as classification scores are used to jointly build a metric as the criteria to assign labels. However, the introduced object branch extends the number of tasks to be aligned from two to three, which obviously increases the difficulty. Based on the experimental results and this analysis, the object loss is then discarded in YOLOv6.

### B.4 EXPERIMENTAL SETUP

We use the same optimizer and the learning schedule as YOLOv5 (Glenn, 2022), i.e., stochastic gradient descent (SGD) with momentum and cosine decay on the learning rate. Warm-up, grouped weight decay strategy and the exponential moving average (EMA) are also utilized. We adopt two strong data augmentations (Mosaic (Bochkovskiy et al., 2020; Glenn, 2022) and Mixup (Zhang et al., 2017a)) following (Bochkovskiy et al., 2020; Glenn, 2022; Ge et al., 2021b). A complete list of hyperparameter settings can be found in our released code. We train our models on the COCO 2017 (Lin et al., 2014) training set, and the accuracy is evaluated on the COCO 2017 validation set. All our models are trained on 8 NVIDIA A100 GPUs, and the speed performance is measured on an NVIDIA Tesla T4 GPU with TensorRT version 7.2 unless otherwise stated. All our models are trained for 300 epochs without pre-training or any external data.

## C DETAILED LATENCY AND THROUGHPUT BENCHMARK

Unless otherwise stated, all the reported latency is measured on an NVIDIA Tesla T4 GPU with TensorRT version 7.2.1.6. Due to the large variance of the hardware and software settings, we re-measure the latency and throughput of all the models under the same configuration (both hardware and software). For a handy reference, we also switch TensorRT versions (Table 20) for consistency check. Latency on a V100 GPU (Table 21) is included for convenient comparison. This gives us a full spectrum view of state-of-the-art detectors. Comparisons about inference speed of YOLOv6 with TensorRT 8.2 on T4 GPU are shown in Table 20. From the table, we can see the throughput of YOLOv6 models still emulates their peers.

### C.1 V100 GPU LATENCY TABLE

Throughout and latency on V100 GPU can be seen in Table 21. And the speed advantage of YOLOv6 is largely maintained.

### C.2 CPU LATENCY

We evaluate the performance of our models and other competitors on a 2.6 GHz Intel Core i7 CPU using OpenCV Deep Neural Network (DNN), as shown in Table 22. Considering the actual application scenario, we only compare the relatively small models (i.e., YOLOv6-N/S/M) with other competitors of save scales.

| Method | FPS (bs=1) | FPS (bs=32) | Latency (bs=1) |
|---|---|---|---|
| YOLOv5-N Glenn (2022) | 702 | 843 | 1.4 ms |
| YOLOv5-S Glenn (2022) | 433 | 515 | 2.3 ms |
| YOLOv5-M Glenn (2022) | 202 | 235 | 4.9 ms |
| YOLOv5-L Glenn (2022) | 126 | 137 | 7.9 ms |
| YOLOX-Tiny Ge et al. (2021b) | 766 | 1393 | 1.3 ms |
| YOLOX-S Ge et al. (2021b) | 313 | 489 | 2.6 ms |
| YOLOX-M Ge et al. (2021b) | 159 | 204 | 5.3 ms |
| YOLOX-L Ge et al. (2021b) | 104 | 117 | 9.0 ms |
| PPYOLOE-S Xu et al. (2022) | 357 | 493 | 2.8 ms |
| PPYOLOE-M Xu et al. (2022) | 163 | 210 | 6.1 ms |
| PPYOLOE-L Xu et al. (2022) | 110 | 145 | 9.1 ms |
| YOLOv7-Tiny Wang et al. (2022) | 464 | 568 | 2.1 ms |
| YOLOv7 Wang et al. (2022) | 128 | 135 | 7.6 ms |
| YOLOv6-N | 785 | 1215 | 1.3 ms |
| YOLOv6-S | 345 | 498 | 2.9 ms |
| YOLOv6-M | 178 | 238 | 5.6 ms |
| YOLOv6-L | 105 | 125 | 9.5 ms |

Table 20: YOLO-series comparison of latency and throughput on a T4 GPU with a higher version of TensorRT (8.2).

| Method | FPS (bs=1) | FPS (bs=32) | Latency (bs=1) |
|---|---|---|---|
| YOLOv5-N Glenn (2022) | 577 | 1727 | 1.4 ms |
| YOLOv5-S Glenn (2022) | 449 | 1249 | 1.7 ms |
| YOLOv5-M Glenn (2022) | 271 | 698 | 3.0 ms |
| YOLOv5-L Glenn (2022) | 178 | 440 | 4.7 ms |
| YOLOX-Tiny Ge et al. (2021b) | 569 | 2883 | 1.4 ms |
| YOLOX-S Ge et al. (2021b) | 386 | 1206 | 2.0 ms |
| YOLOX-M Ge et al. (2021b) | 245 | 600 | 3.4 ms |
| YOLOX-L Ge et al. (2021b) | 149 | 361 | 5.6 ms |
| PPYOLOE-S Xu et al. (2022) | 322 | 1050 | 2.4 ms |
| PPYOLOE-M Xu et al. (2022) | 222 | 566 | 4.0 ms |
| PPYOLOE-L Xu et al. (2022) | 153 | 406 | 5.5 ms |
| YOLOv7-Tiny Wang et al. (2022) | 453 | 1565 | 1.7 ms |
| YOLOv7 Wang et al. (2022) | 182 | 412 | 4.6 ms |
| YOLOv6-N | 646 | 2660 | 1.2 ms |
| YOLOv6-S | 399 | 1330 | 2.0 ms |
| YOLOv6-M | 203 | 676 | 4.4 ms |
| YOLOv6-L | 125 | 385 | 6.8 ms |

Table 21: YOLO-series comparison of latency and throughput on a V100 GPU. We measure all models at FP16-precision with the input size 640×640 in the exact same environment.

| Method | Input | Latency (bs=1) |
|---|---|---|
| YOLOv5-N Glenn (2022) | 640 | 118.9 ms |
| YOLOv5-S Glenn (2022) | 640 | 202.2 ms |
| YOLOX-Tiny Ge et al. (2021b) | 416 | 144.2 ms |
| YOLOX-S Ge et al. (2021b) | 640 | 164.6 ms |
| YOLOX-M Ge et al. (2021b) | 640 | 357.9 ms |
| YOLOv7-Tiny Wang et al. (2022) | 640 | 137.5 ms |
| YOLOv6-N | 640 | 60.3 ms |
| YOLOv6-S | 640 | 148.0 ms |
| YOLOv6-M | 640 | 269.3 ms |

Table 22: YOLO-series comparison of latency on a typical CPU. We measure all models at FP32-precision with the input size $640 \times 640$ in the exact same environment.

