# OpenReview forum: "YOLOV6: A SINGLE-STAGE OBJECT DETECTION FRAMEWORK FOR INDUSTRIAL APPLICATIONS"
_ICLR.cc/2024/Conference — ICLR 2024 Conference Withdrawn Submission_

### Official Review · Reviewer_W4rd · 2023-10-28

**Soundness:** 2 fair
**Presentation:** 2 fair
**Contribution:** 2 fair
**Rating:** 3
**Confidence:** 5

**Summary:**

This paper proposed YOLOv6, a real-time object detector that incorporates a group of modifications to the basic building block and training techniques. The experimental results show that the YOLOv6 obtains a better speed-accuracy trade-off on the TensorRT platform than previous methods and verify the modifications are effective.

**Strengths:**

1. The final performance on the COCO detection benchmark looks good, which makes YOLOv6 an off-the-shelf real-time object detector that the community can widely adopt.
2. The ablation study verifies the effectiveness of each modification.

**Weaknesses:**

1. Limited insights or knowledge are provided in the paper. In the methodology part, the most common pattern of the paper writing is 'the previous work have done A, B, C, and D. Inspired by these, we do xxx' (e.g., in the section of backbone and neck). However, the reviewer cannot get what is the exact inspiration and what is the real motivation and analysis behind the modifications. The reviewer acknowledges the changes that could bring some gains, but why these modifications are done, the principles of these modifications, and the implications of the experimental results are not given, which provides limited insights to benefit the community and makes this paper not appropriate for this venue.
2. The comparison might not be comprehensive. For example, the paper compares with RTMDet on real-time instance segmentation, but RTMDet also provides real-time object detection results. Recent real-time instance segmentation models like SparseInst and SOLOv2 are also not included.

**Questions:**

N/A

---

### Official Review · Reviewer_jptY · 2023-10-30

**Soundness:** 1 poor
**Presentation:** 2 fair
**Contribution:** 2 fair
**Rating:** 3
**Confidence:** 4

**Summary:**

This paper presents a new object detector YOLOv6 for real-time object detection. The technical novelties consist of a new network design, an anchor-based test-free auxiliary branch for training, and an adaptive self-distillation for transferring knowledge from a teacher to a student. Experimental results show the state-of-the-art performance of the proposed YOLOv6 among various YOLO series.

**Strengths:**

1. This paper establishes a real-time object detection framework and achieves state-of-the-art performance on COCO minival 2017.

2. The proposed methods are technically sound.

**Weaknesses:**

1. While the proposed method achieves state-of-the-art performance on COCO val2017, the paper does not show the results on the COCO test-dev set. Since this detector is made for pursuing SoTA performance in object detection, it should provide the evaluation on the COCO test-dev set.

2. The paper introduces several strategies for improving detection performance, some of which are not cost-free in training, e.g., anchor-aided training and self-distillation. However, the paper does not provide the results for training time cost.

3. The comparison is unfair. The self-distillation requires a pre-trained teacher model, which makes the YOLOv6 need double the training time. In contrast, the previous works like YOLOv5, YOLOv8, and DETR series do not need that. I noticed that the paper conducts an experiment to check the performance of double training epochs, showing the superiority of self-distillation. However, in the benchmark test, all the detectors train for 300 epochs, while YOLOv6 actually trains for 600 epochs (300 for teacher and 300 for student). This is a serious problem which makes the comparison unfair.

4. It is shown in Table 17 that SIoU or CIoU loss works better than GIoU. Then why did you choose a sub-optimal one GIoU?

5. The label assignment algorithm TAL, VFL loss, and GIoU loss mentioned in Sec. 4.3.1 lack references in the main body of the paper, though they appear in the Appendix.

6. The cosine weight decay sounds like a regularization while it is not in the proposed self-distillation method. I suggest the author choose a better name for it.

7. There are some inappropriate statements and ambiguities that may be misleading to readers. 1) In Sec. 3.3, you mentioned: "our large models (i.e., YOLOv6-M/L) adopt DFL [1] **as regression loss** for the convenience of **performing self-distillation** on localization". As a matter of fact, DFL is not a box regression loss. It should be the IoU-based losses. DFL loss is a weighted cross-entropy loss, which is proposed as additional supervision to make the convergence faster and more stable. 2) DFL is not originally proposed for distillation. It should be the localization distillation [2]. 3) On the last paragraph of page 5, "Notably, the introduction of DFL (Li et al., 2020) requires extra parameters for the regression branch, ..." DFL (distribution focal loss) is just a loss function. It should be the general distribution representation of the bounding box. 4) What is the experimental setting of double epochs in Table 10?


[1] Li X, Wang W, Wu L, et al. Generalized focal loss: Learning qualified and distributed bounding boxes for dense object detection[J]. Advances in Neural Information Processing Systems, 2020, 33: 21002-21012.

[2] Zheng Z, Ye R, Hou Q, et al. Localization distillation for object detection[J]. IEEE Transactions on Pattern Analysis and Machine Intelligence, 2023.

**Questions:**

see weaknesses

---

### Official Review · Reviewer_3aeD · 2023-10-30

**Soundness:** 3 good
**Presentation:** 3 good
**Contribution:** 2 fair
**Rating:** 3
**Confidence:** 5

**Summary:**

This paper presents a comprehensive set of architectural enhancements and training techniques aimed at refining the YOLO framework, resulting in the creation of YOLOv6, an advanced detection approach tailored for industrial applications. The primary highlights of this work revolve around two innovative training strategies, namely anchor-assisted training and adaptive distillation, in addition to the incorporation of reparameterization techniques to optimize architectural blocks for enhanced inference efficiency. Notably, YOLOv6 exhibits superior performance when evaluated on the COCO dataset, and its efficacy is further validated through a series of ablation studies. This research contributes to the evolution of the YOLO framework for practical industrial settings.

**Strengths:**

Amongst the positive aspects of the paper are the following:

- This paper provides a comprehensive analysis of its proposed methodologies through a series of ablation studies, thereby establishing their efficacy.
- These methodologies are evaluated on the widely recognized COCO evaluation dataset, a standard benchmark for this particular task.
- Notably, the AAT and BiC techniques consistently enhance the model's performance.
- Moreover, the paper demonstrates the potential of transfer learning in two distinct downstream tasks, thereby extending the applicability of the proposed approaches.

**Weaknesses:**

- The reparameterization approach used from (Ding et al., 2021) while effective, may not be attributed as a main contribution of the paper as it merely adapts the main blocks using it, rather than constituting a fundamentally novel contribution.
- Overall, even if AP improvements are observed they are in the range of around 1-2%. This might not be significant and in many cases and settings other methods outperform YOLOv6. The effectiveness of the proposed approaches overall is mixed. Some perform well while others depend on the model and setting. Consequently, the outcomes remain inconclusive.
- The title and introduction of this paper suggest that YOLOv6 is specifically designed for industrial applications. However, the paper falls short in providing a comprehensive exploration of the requirements and specific challenges within the industrial domain. To enhance the quality of this review, it is essential to delve deeper into these aspects.
- The paper introduces several techniques for object detection, which, upon closer examination, do not exhibit a clear specialization for industrial applications. While the paper may present effective object detection strategies, it is crucial to emphasize that these methods are already widely employed in various object detection scenarios. The lack of distinctive features or adjustments tailored specifically to industrial settings raises questions about the paper's unique contribution.

**Questions:**

In summary, while the paper introduces valuable object detection techniques, it falls short of delivering a truly specialized solution for industrial applications. To strengthen its claim of tailor-made suitability for this domain, a more detailed analysis of the specific industrial requirements and challenges, alongside innovative adjustments, is necessary. This would ensure a clearer and more significant contribution to the field of industrial object detection.